# Towards Deep Learning Models Resistant to Adversarial Attacks

**Aleksander Mądry, Aleksandar Makelov, Ludwig Schmidt, Dimitris Tsipras, Adrian Vladu**[*]
Department of Electrical Engineering and Computer Science
Massachusetts Institute of Technology
Cambridge, MA 02139, USA
`{madry,amakelov,ludwigs,tsipras,avladu}@mit.edu`

## Abstract

Recent work has demonstrated that neural networks are vulnerable to adversarial examples, i.e., inputs that are almost indistinguishable from natural data and yet classified incorrectly by the network. To address this problem, we study the adversarial robustness of neural networks through the lens of robust optimization. This approach provides us with a broad and unifying view on much prior work on this topic. Its principled nature also enables us to identify methods for both training and attacking neural networks that are reliable and, in a certain sense, universal. In particular, they specify a concrete security guarantee that would protect against a well-defined class of adversaries. These methods let us train networks with significantly improved resistance to a wide range of adversarial attacks. They also suggest robustness against a *first-order adversary* as a natural security guarantee. We believe that robustness against such well-defined classes of adversaries is an important stepping stone towards fully resistant deep learning models.

## 1 Introduction

Recent breakthroughs in computer vision and speech recognition are bringing trained classifiers into the center of security-critical systems. Important examples include vision for autonomous cars, face recognition, and malware detection. These developments make security aspects of machine learning increasingly important. In particular, resistance to *adversarially chosen inputs* is becoming a crucial design goal. While trained models tend to be very effective in classifying benign inputs, recent work (Dalvi et al., 2004; Szegedy et al., 2013; Goodfellow et al., 2014; Nguyen et al., 2015; Sharif et al., 2016) shows that an adversary is often able to manipulate the input so that the model produces an incorrect output.

This phenomenon has received particular attention in the context of deep neural networks, and there is now a quickly growing body of work on this topic (Fawzi et al., 2015; Kurakin et al., 2016; Papernot & McDaniel, 2016; Rozsa et al., 2016; Torkamani, 2016; Sokolic et al., 2016; Tramèr et al., 2017b). Computer vision presents a particularly striking challenge: very small changes to the input image can fool state-of-the-art neural networks with high probability (Szegedy et al., 2013; Goodfellow et al., 2014; Nguyen et al., 2015; Sharif et al., 2016; Moosavi-Dezfooli et al., 2016). This holds even when the benign example was classified correctly, and the change is imperceptible to a human. Apart from the security implications, this phenomenon also demonstrates that our current models are not learning the underlying concepts in a robust manner. All these findings raise a fundamental question:

*How can we learn models robust to adversarial inputs?*

There are now many proposed defense mechanisms for the adversarial setting. Examples include defensive distillation (Papernot et al., 2016a; Papernot & McDaniel, 2016), feature squeezing (Xu et al., 2017), and several detection approaches for adversarial inputs (see Carlini & Wagner (2017) for references). While these works constitute important first steps in exploring the realm of possibilities, they do not offer a good understanding of the *guarantees* they provide. We can never be certain

---

[*]Authors ordered alphabetically.

that a particular defense mechanism prevents the existence of some well-defined *class* of adversarial attacks. This makes it difficult to navigate the landscape of adversarial robustness or to fully evaluate the possible security implications. Moreover, subsequent work (Carlini & Wagner, 2016a; He et al., 2017) has shown that most of these defenses can be bypassed by stronger, adaptive adversaries.

In this paper, we study the adversarial robustness of neural networks through the lens of robust optimization. We use a natural saddle point (min-max) formulation to capture the notion of security against adversarial attacks in a principled manner. This formulation allows us to be precise about the type of security *guarantee* we would like to achieve, i.e., the broad *class* of attacks we want to be resistant to (in contrast to defending only against specific known attacks). The formulation also enables us to cast both *attacks* and *defenses* into a common theoretical framework. Most prior work on adversarial examples naturally fits into this framework. In particular, adversarial training directly corresponds to optimizing this saddle point problem. Similarly, prior methods for attacking neural networks correspond to specific algorithms for solving the underlying optimization problem.

Equipped with this perspective, we make the following contributions.

1. We conduct a careful experimental study of the optimization landscape corresponding to this saddle point formulation. Despite the non-convexity and non-concavity of its constituent parts, we find that the underlying optimization problem *is* tractable after all. In particular, we provide strong evidence that first-order methods can reliably solve this problem and motivate projected gradient descent (PGD) as a universal "first-order adversary", i.e., the strongest attack utilizing the local first order information about the network. We supplement these insights with ideas from real analysis to further motivate adversarial training against a PGD adversary as a strong and natural defense.

2. We explore the impact of network architecture on adversarial robustness and find that model capacity plays an important role. To reliably withstand strong adversarial attacks, networks require a significantly larger capacity than for correctly classifying benign examples only. This shows that a robust decision boundary of the saddle point problem can be significantly more complicated than a decision boundary that simply separates the benign data points.

3. Building on the above insights, we train networks on MNIST and CIFAR10 that are robust to a wide range of adversarial attacks against adversaries bounded by $0.3$ and $8$ in $\ell_\infty$ norm respectively. Our approach is based on optimizing the aforementioned saddle point formulation and uses our optimal "first-order adversary". Our best MNIST model achieves an accuracy of more than 89% against the strongest adversaries in our test suite. In particular, our MNIST network is even robust against *white box* attacks of an *iterative* adversary. Our CIFAR10 model achieves an accuracy of 46% against the same adversary. Furthermore, in case of the weaker *black box (transfer)* attacks, our MNIST and CIFAR10 networks achieve an accuracy of more than 95% and 64%, respectively (a more detailed overview can be found in Tables 1 and 2). To the best of our knowledge, we are the first to achieve these levels of robustness on MNIST and CIFAR10 against a broad set of attacks.

Overall, these findings suggest that secure neural networks are within reach. In order to further support this claim, we have invited the community to attempt attacks against our MNIST and CIFAR10 networks in the form of an open challenge[1,2]. At the time of writing, we received about fifteen submissions to the MNIST challenge and the best submission achieved roughly 93% accuracy in a black box attack. We received no submissions for the CIFAR10 challenge that went beyond the 64% accuracy of our attack. Considering that other proposed defenses were often quickly broken (Carlini & Wagner, 2017), we believe that our robust models are significant progress on the defense side. Furthermore, recent work (Carlini et al., 2017) on verifiable adversarial examples showed that our proposed defense reliably increased the robustness to *any* $\ell_\infty$-bounded attack.

## 2 AN OPTIMIZATION VIEW ON ADVERSARIAL ROBUSTNESS

Much of our discussion will revolve around an optimization view of adversarial robustness. This perspective not only captures the phenomena we want to study in a precise manner, but will also

---

[1]https://github.com/MadryLab/mnist_challenge
[2]https://github.com/MadryLab/cifar10_challenge

inform our investigations. To this end, let us consider a standard classification task with an underlying data distribution $\mathcal{D}$ over pairs of examples $x \in \mathbb{R}^d$ and corresponding labels $y \in [k]$. We also assume that we are given a suitable loss function $L(\theta, x, y)$, for instance the cross-entropy loss for a neural network. As usual, $\theta \in \mathbb{R}^p$ is the set of model parameters. Our goal then is to find model parameters $\theta$ that minimize the risk $\mathbb{E}_{(x,y) \sim \mathcal{D}}[L(x, y, \theta)]$.

Empirical risk minimization (ERM) has been tremendously successful as a recipe for finding classifiers with small population risk. Unfortunately, ERM often does not yield models that are robust to adversarially crafted examples (Goodfellow et al., 2014; Kurakin et al., 2016; Moosavi-Dezfooli et al., 2016; Tramèr et al., 2017b). Formally, there are efficient algorithms ("adversaries") that take an example $x$ belonging to class $c_1$ as input and find examples $x^{\text{adv}}$ such that $x^{\text{adv}}$ is very close to $x$ but the model incorrectly classifies $x^{\text{adv}}$ as belonging to class $c_2 \neq c_1$.

In order to *reliably* train models that are robust to adversarial attacks, it is necessary to augment the ERM paradigm. Instead of resorting to methods that directly focus on improving the robustness to specific attacks, our approach is to first propose a concrete *guarantee* that an adversarially robust model should satisfy. We then adapt our training methods towards achieving this guarantee.

The first step towards such a guarantee is to specify an *threat model*, i.e., a precise definition of the attacks our models should be resistant to. For each data point $x$, we introduce a set of allowed perturbations $\mathcal{S} \subseteq \mathbb{R}^d$ that formalizes the manipulative power of the adversary. In image classification, we choose $\mathcal{S}$ so that it captures perceptual similarity between images. For instance, the $\ell_\infty$-ball around $x$ has recently been studied as a natural notion for adversarial perturbations (Goodfellow et al., 2014). While we focus on robustness against $\ell_\infty$-bounded attacks in this paper, we remark that more comprehensive notions of perceptual similarity are an important direction for future research.

Next, we modify the definition of population risk $\mathbb{E}_\mathcal{D}[L]$ by incorporating the above adversary. Instead of computing the loss $L$ directly on samples from the distribution $\mathcal{D}$, we allow the adversary to perturb the input first. This gives rise to the following saddle point problem, which is our central object of study:

$$\min_\theta \rho(\theta), \quad \text{where} \quad \rho(\theta) = \mathbb{E}_{(x,y) \sim \mathcal{D}} \left[ \max_{\delta \in \mathcal{S}} L(\theta, x + \delta, y) \right] . \tag{2.1}$$

Formulations of this type (and their finite-sample counterparts) have a long history in robust optimization, going back to Wald (Wald, 1939; 1945; 1992). It turns out that this formulation is also particularly useful in our context. We will refer to the quantity $\rho(\theta)$ as the *adversarial loss* of the network with parameters $\theta$.

First, this formulation gives us a unifying perspective that encompasses much prior work on adversarial robustness. Our perspective stems from viewing the saddle point problem as the composition of an *inner maximization* problem and an *outer minimization* problem. Both of these problems have a natural interpretation in our context. The inner maximization problem aims to find an adversarial version of a given data point $x$ that achieves a high loss. This is precisely the problem of attacking a given neural network. On the other hand, the goal of the outer minimization problem is to find model parameters so that the adversarial loss given by the inner attack problem is minimized. This is precisely the problem of training a robust classifier using adversarial training techniques.

Second, the saddle point problem specifies a clear goal that a robust classifier should achieve, as well as a quantitative measure of its robustness. In particular, when the parameters $\theta$ yield a (nearly) vanishing risk, the corresponding model is perfectly robust to attacks specified by our threat model.

Our paper investigates the structure of this saddle point problem in the context of deep neural networks. This formulation will be the main drive of our investigations that will lead us to training techniques that produce models with high resistance to a wide range of adversarial attacks.

# 3 TOWARDS ADVERSARIALLY ROBUST NETWORKS

Current work on adversarial examples usually focuses on specific defensive mechanisms, or on attacks against such defenses. An important feature of formulation (2.1) is that attaining small adversarial loss gives a *guarantee* that no allowed attack will fool the network. By definition, no adversarial perturbations are possible because the loss is small for *all* perturbations allowed by our

threat model. This perspective allows us to reduce the task of finding truly robust models to an optimization problem. Hence, we can now focus our attention solely on obtaining a good solution to Problem (2.1).

**Gradients from attacks.** Since Stochastic Gradient Descent (SGD) and its variants are by far the most successful algorithms for training neural networks, we also want to apply SGD to Problem (2.1). This raises the question how we can compute gradients $\nabla_\theta \rho(\theta)$ for the outer minimization problem. Since the adversarial loss function $\rho(\theta)$ corresponds to a maximization problem, we cannot simply apply the usual backpropagation algorithm. Instead, a natural approach is to compute the gradient at the *maximizer* of the inner maximization problem. A priori, it is not clear that this is a valid descent direction for the saddle point problem. However, for the case of continuously differentiable functions, Danskin's theorem – a classic theorem in optimization – states that this is indeed true and gradients at maximizers of the inner problem correspond to descent directions for the saddle point problem (see Appendix C for details).

Leveraging this connection, our goal now is to find a reliable algorithm for solving the inner maximization problem, i.e., to evaluate $\rho(\theta)$. When instantiated for a batch of examples (instead of the expectation over the entire distribution $\mathcal{D}$), finding a maximizer $\delta \in \mathcal{S}$ of $\rho(\theta)$ corresponds exactly to finding an attack on the neural network. This allows us to employ known attacks as inner maximization algorithms. Prior work has proposed methods such as the Fast Gradient Sign Method (FGSM) and multiple variations of it (Goodfellow et al., 2014). FGSM is an attack for an $\ell_\infty$-bounded adversary and computes an adversarial example as

$$x + \varepsilon \operatorname{sgn}(\nabla_x L(\theta, x, y)).$$

One can interpret this attack as a simple one-step scheme for maximizing the inner part of the saddle point formulation. A more powerful adversary is the multi-step variant FGSM$^k$, which is essentially projected gradient descent (PGD) on the negative loss function (Kurakin et al., 2016):[3]

$$x^{t+1} = \operatorname{Proj}_{x+\mathcal{S}} \left( x^t + \alpha \operatorname{sgn}(\nabla_{x^t} L(\theta, x^t, y)) \right).$$

**Loss landscape.** While PGD is a well-motivated approach for the inner maximization problem, it is not clear whether we can actually find a good solution in a reasonable amount of time. The problem is non-concave, so a priori we have no guarantees on the solution quality of PGD. One of our contributions is demonstrating that, in practice, the inner maximization problem is indeed well-behaved. In particular, we experimentally explore the structure given by the non-concave inner problem and find that its loss landscape has a surprisingly tractable structure of local maxima (see Appendix A). This structure also points towards projected gradient descent as the "ultimate" first-order adversary (see Section 5).

Despite the fact that the exact assumptions of Danskin's theorem do not hold for our problem (the function is not continuously differentiable due to ReLU activations, and we only compute approximate maximizers of the inner problem), our experiments suggest that we can still use these gradients to optimize our problem. By applying SGD using the gradient of the loss at adversarial examples, we can consistently reduce the loss of the saddle point problem during training (e.g., see Figure 1 in Section 4). These observations suggest that we reliably optimize the saddle point formulation (2.1) and thus train robust classifiers.

**Model capacity.** Before we proceed to our main experiment results in the next section, we briefly mention another important insight from our robust optimization perspective. Solving the problem from Equation (2.1) successfully is not sufficient to guarantee robust and accurate classification. We also require that the *value* of the problem (i.e., the final loss we achieve against adversarial examples) is small, which then provides guarantees for the performance of our classifier. In particular, achieving a very small value corresponds to a perfect classifier, which is robust to adversarial inputs. In Appendix B, we show experimentally that network capacity plays a crucial role in enabling robustness. In particular, training a robust classifier requires a significantly larger network than only achieving high accuracy on natural examples.

---

[3]Other methods like FGSM with random perturbation have also been proposed (Tramèr et al., 2017a). All of these approaches can be seen as specific attempts to solve the inner maximization problem in (2.1).

## 4 EXPERIMENTS: ADVERSARIALLY ROBUST DEEP LEARNING MODELS?

Following our understanding developed in the previous section, we can now apply our proposed approach to train robust classifiers. For both MNIST and CIFAR10, our adversary of choice will be projected gradient descent starting from a random perturbation around the natural example. As our experiments suggest (Appendix A) this algorithm is very efficient at reliably producing examples of (near) maximal loss. In a sense, it seems to correspond to a "ultimate" first order adversary. Since we are training the model for multiple epochs, we did not see any benefit in restarting PGD multiple times per batch – a new start is chosen each time the same example is encountered.

During the training procedure against the PGD adversary, we observe a steady decrease in the training loss of adversarial examples, illustrated in Figure 1. This behavior indicates that we are consistently decreasing the adversarial loss and indeed successfully solving our original optimization problem.

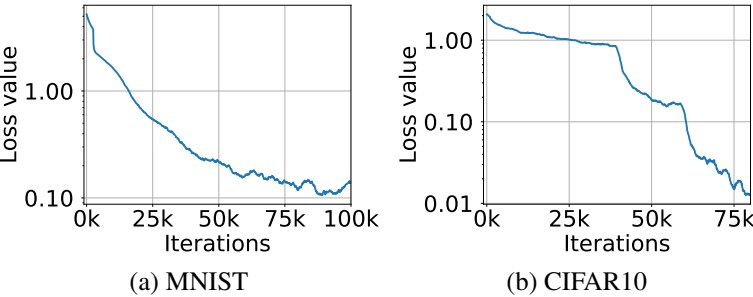

(a) MNIST            (b) CIFAR10

Figure 1: Cross-entropy loss on adversarial examples during training. The plots show how the adversarial loss on training examples evolves during training the MNIST and CIFAR10 networks against a PGD adversary. The sharp drops in the CIFAR10 plot correspond to decreases in training learning rate. These plots illustrate that we can consistently reduce the value of the inner problem of the saddle point formulation (2.1), thus producing an increasingly robust classifier.

We evaluate the trained models against a range of adversaries. We illustrate our results in Table 1 for MNIST and Table 2 for CIFAR10. The adversaries we consider are:

- White-box attacks with PGD for a different number of of iterations and restarts, denoted by source A.
- White-box attacks from Carlini & Wagner (2016b). We use their suggested loss function and minimize it using PGD. This is denoted as CW, where the corresponding attack with a high confidence parameter ($\kappa = 50$) is denoted as CW+.
- Black-box attacks from an independently trained copy of the network, denoted A'.
- Black-box attacks from a version of the same network trained only on natural examples, denoted $A_{nat}$.
- Black-box attacks from a different convolution architecture, denoted B, described in Tramèr et al. (2017a).

**MNIST.**  We run 40 iterations of projected gradient descent as our adversary, with a step size of 0.01 (we choose to take gradient steps in the $\ell_\infty$ norm, i.e. adding the sign of the gradient, since this makes the choice of the step size simpler). We train and evaluate against perturbations of size $\varepsilon = 0.3$. We use a network consisting of two convolutional layers with 32 and 64 filters respectively, each followed by $2 \times 2$ max-pooling, and a fully connected layer of size 1024. When trained with natural examples, this network reaches 99.2% accuracy on the evaluation set. However, when evaluating on examples perturbed with FGSM the accuracy drops to 6.4%. Given that the resulting MNIST model is very robust, we investigated the learned parameters in order to understand how they affect adversarial robustness. The results of the investigation are presented in Appendix E.

**CIFAR10.**  For the CIFAR10 dataset, we use the two architectures described in B (the original Resnet and its $10\times$ wider variant). We trained the network against a PGD adversary with $\ell_\infty$ projected

Table 1: MNIST: Performance of the adversarially trained network against different adversaries for $\varepsilon = 0.3$. For each model of attack we show the most successful attack with bold. The source networks used for the attack are: the network itself (A) (white-box attack), an indepentenly initialized and trained copy of the network (A'), architecture B from Tramèr et al. (2017a) (B).

| Method | Steps | Restarts | Source | Accuracy |
|--------|-------|----------|--------|----------|
| Natural | - | - | - | 98.8% |
| FGSM | - | - | A | 95.6% |
| PGD | 40 | 1 | A | 93.2% |
| PGD | 100 | 1 | A | 91.8% |
| PGD | 40 | 20 | A | 90.4% |
| PGD | 100 | 20 | A | **89.3%** |
| Targeted | 40 | 1 | A | 92.7% |
| CW | 40 | 1 | A | 94.0% |
| CW+ | 40 | 1 | A | 93.9% |
| FGSM | - | - | A' | 96.8% |
| PGD | 40 | 1 | A' | 96.0% |
| PGD | 100 | 20 | A' | **95.7%** |
| CW | 40 | 1 | A' | 97.0% |
| CW+ | 40 | 1 | A' | 96.4% |
| FGSM | - | - | B | **95.4%** |
| PGD | 40 | 1 | B | 96.4% |
| CW+ | - | - | B | 95.7% |

gradient descent again, this time using 7 steps of size 2, and a total $\varepsilon = 8$. For our hardest adversary we chose 20 steps with the same settings, since other hyperparameter choices didn't offer a significant decrease in accuracy. The results of our experiments appear in Table 2. The adversarial robustness of our network is significant, given the power of iterative adversaries, but still far from satisfactory. We believe that further progress is possible along these lines by understanding how adversarial training works and what techniques can complement it leading to robust models.

Table 2: CIFAR10: Performance of the adversarially trained network against different adversaries for $\varepsilon = 8$. For each model of attack we show the most effective attack in bold. The source networks considered for the attack are: the network itself (A) (white-box attack), an independtly initialized and trained copy of the network (A'), a copy of the network trained on natural examples ($A_{nat}$).

| Method | Steps | Source | Accuracy |
|--------|-------|--------|----------|
| Natural | - | - | 87.3% |
| FGSM | - | A | 56.1% |
| PGD | 7 | A | 50.0% |
| PGD | 20 | A | **45.8%** |
| CW | 30 | A | 46.8% |
| FGSM | - | A' | 67.0% |
| PGD | 7 | A' | **64.2%** |
| CW | 30 | A' | 78.7% |
| FGSM | - | $A_{nat}$ | 85.6% |
| PGD | 7 | $A_{nat}$ | 86.0% |

**Resistance for different values of $\varepsilon$ and $\ell_2$-bounded attacks.** In order to perform a broader evaluation of the adversarial robustness of our models, we run two kinds of additional experiments. On one hand, we investigate the resistance to $\ell_\infty$-bounded attacks for different values of $\varepsilon$. On the other hand, we examine the resistance of our model to attacks that are bounded in $\ell_2$ as opposed to $\ell_\infty$ norm. The results appear in Figure 2. We emphasize that the models we are examining here correspond to training against $\ell_\infty$-bounded attacks with the original value of $\varepsilon = 0.3$, for

MNIST, and $\varepsilon = 8$ for CIFAR10. In particular, our MNIST model retains significant resistance to $\ell_2$-norm-bounded perturbations too – it has good accuracy even for $\varepsilon = 4.5$. We provide a sample of corresponding adversarial examples in Figure 12 of Appendix F. One can observe that some of the underlying perturbations are large enough that even a human could be confused.

**Training Accuracy.** It is worth noting our MNIST and (wide) CIFAR10 networks reached 100% adversarial accuracy on the training set. That is we can fit the training set even against a PGD adversary of $\varepsilon = 0.3$ and $\varepsilon = 8$ respectively. This shows that the landscape of the underlying optimization problem is tractable and does not present a significant barrier to our techniques.

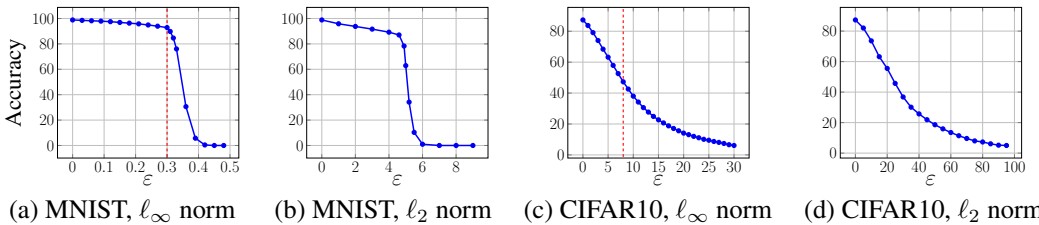

(a) MNIST, $\ell_\infty$ norm     (b) MNIST, $\ell_2$ norm     (c) CIFAR10, $\ell_\infty$ norm     (d) CIFAR10, $\ell_2$ norm

Figure 2: Performance of our adversarially trained networks against PGD adversaries of different strength. The MNIST and CIFAR10 networks were trained against $\varepsilon = 0.3$ and $\varepsilon = 8$ PGD $\ell_\infty$ adversaries respectively (the training $\varepsilon$ is denoted with a red dashed lines in the $\ell_\infty$ plots). We notice that for $\varepsilon$ less or equal to the value used during training, the performance is equal or better.

**Running Time.** Unfortunately, solving the robust version of the problem instead of the standard one imposes a significant computational overhead. Standard training requires one forward and one backward pass through the network for each training batch. Instead, adversarial training with a $k$-step PGD adversary, requires additionally $k$ forward and $k$ backward passes through the network to compute the adversarial version of the training batch. This implies an increase in running time of a factor of $(k + 1)$. We hope that future research will propose ways to mitigate this drawback.

## 5 FIRST-ORDER ADVERSARIES.

Our exploration of the loss landscape (Appendix A) shows that the local maxima found by PGD all have similar loss values, both for normally trained networks and adversarially trained networks. This concentration phenomenon suggests an intriguing view on the problem in which robustness against the PGD adversary yields robustness against *all* first-order adversaries, i.e., attacks that rely only on first-order information. As long as the adversary only uses gradients of the loss function with respect to the input, we conjecture that it will not find significantly better local maxima than PGD. This hypothesis is validated by the experimental evidence provided in Section 4: if we train a network to be robust against PGD adversaries, it becomes robust against a wide range of other attacks as well.

Of course, our exploration with PGD does not preclude the existence of some isolated maxima with much larger function value. However, our experiments suggest that such better local maxima are *hard to find* with first order methods: even a large number of random restarts did not find function values with significantly different loss values (see Appendix A). Incorporating the computational power of the adversary into the threat model should be reminiscent of the notion of *polynomially bounded* adversary that is a cornerstone of modern cryptography. There, this classic threat model allows the adversary to only solve problems that require at most polynomial computation time. Here, we employ an *optimization-based* view on the power of the adversary as it is more suitable in the context of machine learning. After all, we have not yet developed a thorough understanding of the computational complexity of many recent machine learning problems. However, the vast majority of optimization problems in ML is solved with first-order methods, and variants of SGD are the most effective way of training deep learning models in particular. Hence we believe that the class of attacks relying on first-order information is, in some sense, universal for the current practice of deep learning.

Put together, these two ideas chart the way towards machine learning models with *guaranteed* robustness. If we train the network to be robust against PGD adversaries, it will be robust against a wide range of attacks that encompasses all current approaches.

In fact, this robustness guarantee would become even stronger in the context of *transfer attacks*, i.e., attacks in which the adversary does not have a direct access to the target network. Instead, the adversary only has less specific information such as the (rough) model architecture and the training data set. One can view this threat model as an example of "zero order" attacks, i.e., attacks in which the adversary has no direct access to the classifier and is only able to evaluate it on chosen examples without gradient feedback. Still, even for the case of zero-order attacks, the gradient of the network can be estimated using a finite differences method, rendering first-order attacks also relevant in this context.

We discuss transferability in Appendix D. We observe that increasing network capacity and strengthening the adversary we train against (FGSM or PGD training, rather than natural training) improves resistance against transfer attacks. Also, as expected, the resistance of our best models to such attacks tends to be significantly larger than to the (strongest) first order attacks.

# 6 Related Work

Due to the growing body of work on adversarial examples in the context of deep learning networks (Gu & Rigazio, 2014; Fawzi et al., 2015; Torkamani, 2016; Papernot et al., 2016b; Carlini & Wagner, 2016a; Tramèr et al., 2017b; Goodfellow et al., 2014; Kurakin et al., 2016), we focus only on the most related papers here. Before we compare our contributions, we remark that robust optimization has been studied outside deep learning for multiple decades. We refer the reader to Ben-Tal et al. (2009) for an overview of this field.

To the best of our knowledge, in the context of adversarial examples, an explicit formulation of the min-max optimization first appeared in Huang et al. (2015), Shaham et al. (2015), and Lyu et al. (2015). All of these works, however, consider very weak adversaries/methods for solving the maximization problem, mainly relying on linearizing the loss and performing a single step, similar to FGSM. These adversaries do not capture the full range of possible attacks and thus training only against them leaves the resulting classifier vulnerable to more powerful, iterative attacks.

Recent work on adversarial training on ImageNet also observed that the model capacity is important for adversarial training Kurakin et al. (2016). However, their work was focused on FGSM attacks, since they report the iterative attacks are too expensive computationally and don't provide any significant benefits. In contrast to that, we discover that for the datasets we considered training against iterative adversaries *does* result in a model that is robust against such adversaries.

A more recent paper (Tramèr et al., 2017b) also explores the transferability phenomenon. This exploration focuses mostly on the region around natural examples where the loss is (close to) linear. When large perturbations are allowed, this region does not give a complete picture of the adversarial landscape. This is confirmed by our experiments, as well as pointed out by Tramèr et al. (2017a).

Another recent paper (Tramèr et al., 2017a), considers adversarial training using black-box attacks from similar networks in order to increase the robustness of the network against such adversaries. However, this is not an effective defense against the white-box setting we consider, since a PGD adversary can reliably produce adversarial examples for such networks.

# 7 Conclusion

Our findings provide evidence that deep neural networks can be made resistant to adversarial attacks. As our theory and experiments indicate, we can design reliable adversarial training methods. One of the key insights behind this is the unexpectedly regular structure of the underlying optimization task: even though the relevant problem corresponds to the maximization of a highly non-concave function with many distinct local maxima, their *values* are highly concentrated. Overall, our findings give us hope that adversarially robust deep learning models may be within current reach.

For the MNIST dataset, our networks are very robust, achieving high accuracy for a wide range of powerful adversaries and large perturbations. Our experiments on CIFAR10 have not reached the same level of performance yet. However, our results already show that our techniques lead to significant increase in the robustness of the network. We believe that further exploring this direction will lead to adversarially robust networks for this dataset.

ACKNOWLEDGMENTS

Aleksander Mądry, Aleksandar Makelov, and Dimitris Tsipras were supported by the NSF Grant No. 1553428, a Google Research Fellowship, and a Sloan Research Fellowship. Ludwig Schmidt was supported by a Google PhD Fellowship. Adrian Vladu was supported by the NSF Grants No. 1111109 and No. 1553428.

We thank Wojciech Matusik for kindly providing us with computing resources to perform this work.

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

## A    THE LANDSCAPE OF ADVERSARIAL EXAMPLES

The inner problem of the saddle point formulation (2.1) corresponds to finding an adversarial example for a given network and data point (subject to our attack model). As this problem requires us to maximize a highly non-concave function, one would expect it to be intractable. Indeed, this is the conclusion reached by prior work which then resorted to linearizing the inner maximization problem (Huang et al., 2015; Shaham et al., 2015). As pointed out above, this linearization approach yields well-known methods such as FGSM. While training against FGSM adversaries has shown some successes, recent work also highlights important shortcomings of this one-step approach (Tramèr et al., 2017a).

To understand the inner problem in more detail, we investigate the landscape of local maxima for multiple models on MNIST and CIFAR10. The main tool in our experiments is projected gradient descent (PGD), since it is the standard method for large-scale constrained optimization. In order to explore a large part of the loss landscape, we re-start PGD from many points in the $\ell_\infty$ balls around data points from the respective evaluation sets.

Surprisingly, our experiments show that the inner problem *is* tractable after all, at least from the perspective of first-order methods. While there are many local maxima spread widely apart within $x_i + \mathcal{S}$, they tend to have very *well-concentrated* loss *values*. This echoes the folklore belief that training neural networks is possible because the loss (as a function of model parameters) typically has many local minima with very similar values.

Specifically, in our experiments we found the following phenomena:

- We observe that the loss achieved by the adversary increases in a fairly consistent way and plateaus rapidly when performing projected $\ell_\infty$ gradient descent for randomly chosen starting points inside $x + \mathcal{S}$ (see Figure 3).

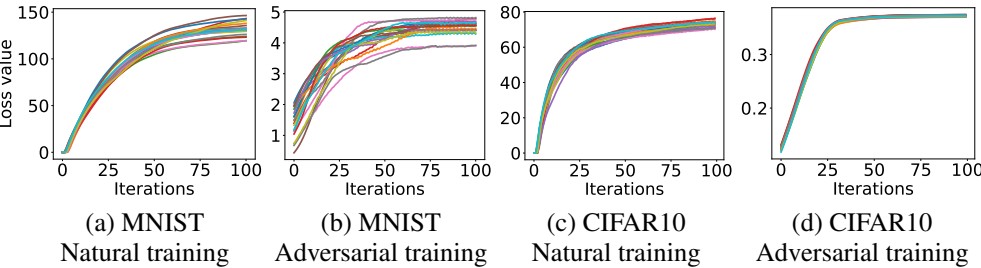

| (a) MNIST | (b) MNIST | (c) CIFAR10 | (d) CIFAR10 |
| Natural training | Adversarial training | Natural training | Adversarial training |

Figure 3: Cross-entropy loss values while creating an adversarial example from the MNIST and CIFAR10 evaluation datasets. The plots show how the loss evolves during 20 runs of projected gradient descent (PGD). Each run starts at a uniformly random point in the $\ell_\infty$-ball around the same natural example (additional plots for different examples appear in Figure 11). The adversarial loss plateaus after a small number of iterations. The optimization trajectories and final loss values are also fairly clustered, especially on CIFAR10. Moreover, the final loss values on adversarially trained networks are significantly smaller than on their naturally trained counterparts.

- Investigating the concentration of maxima further, we observe that over a large number of random restarts, the loss of the final iterate follows a well-concentrated distribution without extreme outliers (see Figure 4; we verified this concentration based on $10^5$ restarts).

- To demonstrate that maxima are noticeably distinct, we also measured the $\ell_2$ distance and angles between all pairs of them and observed that distances are distributed close to the expected distance between two random points in the $\ell_\infty$ ball, and angles are close to $90°$. Along the line segment between local maxima, the loss is convex, attaining its maximum at the endpoints and is reduced by a constant factor in the middle. Nevertheless, for the entire segment, the loss is considerably higher than that of a random point.

- Finally, we observe that the distribution of maxima suggests that the recently developed subspace view of adversarial examples is not fully capturing the richness of attacks (Tramèr et al., 2017b). In particular, we observe adversarial perturbations with negative inner product with the gradient

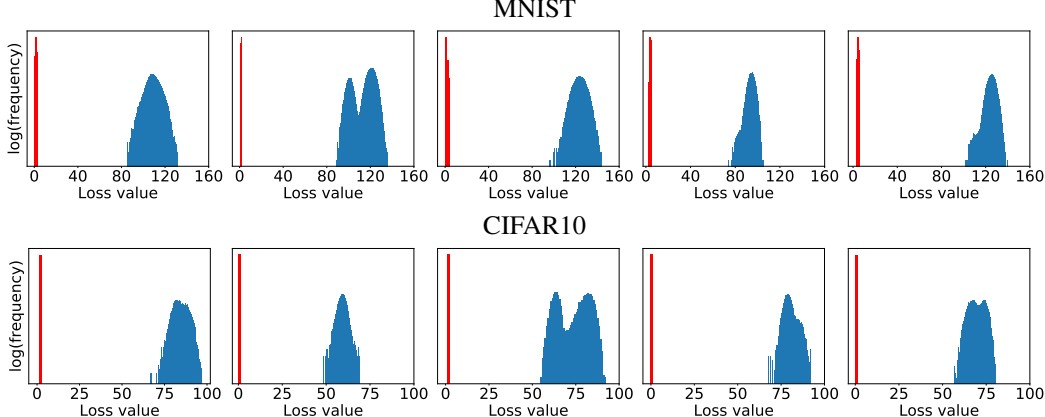

Figure 4: Values of the local maxima given by the cross-entropy loss for five examples from the MNIST and CIFAR10 evaluation datasets. For each example, we start projected gradient descent (PGD) from $10^5$ uniformly random points in the $\ell_\infty$-ball around the example and iterate PGD until the loss plateaus. The blue histogram corresponds to the loss on a naturally trained network, while the red histogram corresponds to the adversarially trained counterpart. The loss is significantly smaller for the adversarially trained networks, and the final loss values are very concentrated without any outliers.

of the example, and deteriorating overall correlation with the gradient direction as the scale of perturbation increases.

## B   NETWORK CAPACITY AND ADVERSARIAL ROBUSTNESS

For a fixed set $\mathcal{S}$ of possible perturbations, the value of the problem (2.1) is entirely dependent on the architecture of the classifier we are learning. Consequently, the architectural capacity of the model becomes a major factor affecting its overall performance. At a high level, classifying examples in a robust way requires a stronger classifier, since the presence of adversarial examples changes the decision boundary of the problem to a more complicated one (see Figure 5 for an illustration).

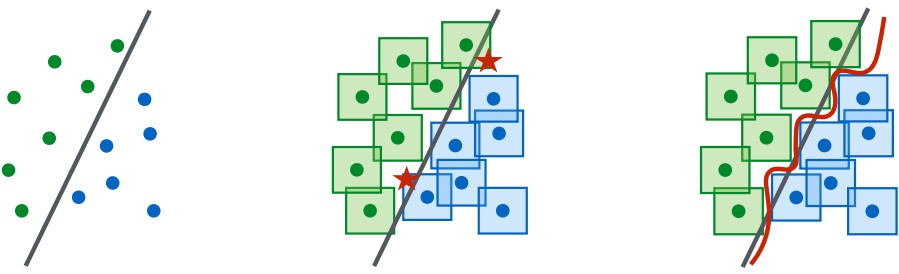

Figure 5: A conceptual illustration of "natural" vs. "adversarial" decision boundaries. Left: A set of points that can be easily separated with a simple (in this case, linear) decision boundary. Middle: The simple decision boundary does not separate the $\ell_\infty$-balls (here, squares) around the data points. Hence there are adversarial examples (the red stars) that will be misclassified. Right: Separating the $\ell_\infty$-balls requires a significantly more complicated decision boundary. The resulting classifier is robust to adversarial examples with bounded $\ell_\infty$-norm perturbations.

Our experiments verify that capacity is crucial for robustness, as well as for the ability to successfully train against strong adversaries. For the MNIST dataset, we consider a simple convolutional network and study how its behavior changes against different adversaries as we keep doubling the size of network (i.e. double the number of convolutional filters and the size of the fully connected layer). The initial network has a convolutional layer with 2 filters, followed by another convolutional layer

with 4 filters, and a fully connected hidden layer with 64 units. Convolutional layers are followed by $2 \times 2$ max-pooling layers and adversarial examples are constructed with $\varepsilon = 0.3$. The results are in Figure 6.

For the CIFAR10 dataset, we used the Resnet model He et al. (2016); TFM (2017). We performed data augmentation using random crops and flips, as well as per image standarization. To increase the capacity, we modified the network incorporating wider layers by a factor of 10. This results in a network with 5 residual units with (16, 160, 320, 640) filters each. This network can achieve an accuracy of 95.2% when trained with natural examples. Adversarial examples were constructed with $\varepsilon = 8$. Results on capacity experiments appear in Figure 6.

We observe the following phenomena:

**Capacity alone helps.** We observe that increasing the capacity of the network when training using only natural examples (apart from increasing accuracy on these examples) increases the robustness against one-step perturbations. This effect is greater when considering adversarial examples with smaller $\varepsilon$.

**FGSM adversaries don't increase robustness (for large $\varepsilon$).** When training the network using adversarial examples generated with the FGSM, we observe that the network overfits to these adversarial examples. This behavior is known as label leaking Kurakin et al. (2016) and stems from the fact that the adversary produces a very restricted set of adversarial examples that the network can overfit to. These networks have poor performance on natural examples and don't exhibit any kind of robustness against PGD adversaries. For the case of smaller $\varepsilon$ the loss is ofter linear enough in the $\ell_\infty$ ball around natural examples, that FGSM finds adversarial examples close to those found by PGD thus being a reasonable adversary to train against.

**Weak models may fail to learn non-trivial classifiers.** In the case of small capacity networks, attempting to train against a strong adversary (PGD) prevents the network from learning anything meaningful. The network converges to always predicting a fixed class, even though it could converge to an accurate classifier through natural training. The small capacity of the network forces the training procedure to sacrifice performance on natural examples in order to provide any kind of robustness against adversarial inputs.

**The value of the saddle point problem decreases as we increase the capacity.** Fixing an adversary model, and training against it, the value of (2.1) drops as capacity increases, indicating the the model can fit the adversarial examples increasingly well.

**More capacity and stronger adversaries decrease transferability.** Either increasing the capacity of the network, or using a stronger method for the inner optimization problem reduces the effectiveness of transferred adversarial inputs. We validate this experimentally by observing that the correlation between gradients from the source and the transfer network, becomes less significant as capacity increases. We describe our experiments in Appendix D.

## C  STATEMENT AND APPLICATION OF DANSKIN'S THEOREM

Recall that our goal is to minimize the value of the saddle point problem

$$\min_\theta \rho(\theta), \quad \text{where} \quad \rho(\theta) = \mathbb{E}_{(x,y)\sim\mathcal{D}} \left[ \max_{\delta\in\mathcal{S}} L(\theta, x + \delta, y) \right] .$$

In practice, we don't have access to the distribution $\mathcal{D}$ so both the gradients and the value of $\rho(\theta)$ will be computed using sampled input points. Therefore we can consider –without loss of generality– the case of a single random example $x$ with label $y$, in which case the problem becomes

$$\min_\theta \max_{\delta\in\mathcal{S}} g(\theta, \delta), \quad \text{where} \quad g(\theta, \delta) = L(\theta, x + \delta, y) .$$

If we assume that the loss $L$ is continuously differentiable in $\theta$, we can compute a descent direction for $\theta$ by utilizing the classical theorem of Danskin.

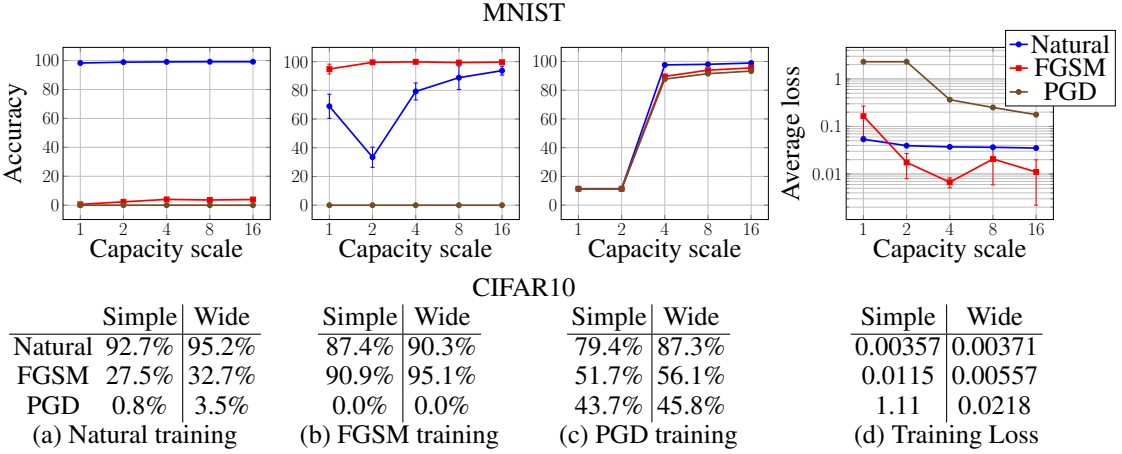

Figure 6: The effect of network capacity on the performance of the network. We trained MNIST and CIFAR10 networks of varying capacity on: (a) natural examples, (b) with FGSM-made adversarial examples, (c) with PGD-made adversarial examples. In the first three plots/tables of each dataset, we show how the natural and adversarial accuracy changes with respect to capacity for each training regime. In the final plot/table, we show the value of the cross-entropy loss on the adversarial examples the networks were trained on. This corresponds to the value of our saddle point formulation (2.1) for different sets of allowed perturbations.

**Theorem C.1** (Danskin). *Let $\mathcal{S}$ be nonempty compact topological space and $g : \mathbb{R}^n \times \mathcal{S} \to \mathbb{R}$ be such that $g(\cdot, \delta)$ is differentiable for every $\delta \in \mathcal{S}$ and $\nabla_\theta g(\theta, \delta)$ is continuous on $\mathbb{R}^n \times \mathcal{S}$. Also, let $\delta^*(\theta) = \{\delta \in \arg\max_{\delta \in \mathcal{S}} g(\theta, \delta)\}$.*

*Then the corresponding max-function*

$$\phi(\theta) = \max_{\delta \in \mathcal{S}} g(\theta, \delta)$$

*is locally Lipschitz continuous, directionally differentiable, and its directional derivatives satisfy*

$$\phi'(\theta, h) = \sup_{\delta \in \delta^*(\theta)} h^\top \nabla_\theta g(\theta, \delta) \,.$$

*In particular, if for some $\theta \in \mathbb{R}^n$ the set $\delta^*(\theta) = \{\delta_\theta^*\}$ is a singleton, the the max-function is differentiable at $\theta$ and*

$$\nabla \phi(\theta) = \nabla_\theta g(\theta, \delta_\theta^*).$$

The intution behind the theorem is that since gradients are local objects, and the function $\phi(\theta)$ is locally the same as $g(\theta, \delta_\theta^*)$ their gradients will be the same. The theorem immediately gives us the following corollary, stating the we can indeed compute gradients for the saddle point by computing gradients at the inner optimizers.

**Corollary C.2.** *Let $\overline{\delta}$ be such that $\overline{\delta} \in \mathcal{S}$ and is a maximizer for $\max_\delta L(\theta, x + \delta, y)$. Then, as long as it is nonzero, $-\nabla_\theta L(\theta, x + \overline{\delta}, y)$ is a descent direction for $\phi(\theta) = \max_{\delta \in \mathcal{S}} L(\theta, x + \delta, y)$.*

*Proof of Corollary C.2.* We apply Theorem C.1 to $g(\theta, \delta) := L(\theta, x + \delta, y)$ and $\mathcal{S} = B_{\|\cdot\|}(\varepsilon)$. We see that the directional derivative in the direction of $h = \nabla_\theta L(\theta, x + \overline{\delta}, y)$ satisfies

$$\phi'(\theta, h) = \sup_{\delta \in \delta^*(\theta)} h^\top \nabla_\theta L(\theta, x + \delta, y) \geq h^\top h = \|\nabla_\theta L(\theta, x + \overline{\delta}, y)\|_2^2 \geq 0 \,.$$

If this gradient is nonzero, then the inequality above is strict. Therefore it gives a descent direction. $\qquad \square$

A technical issue is that, since we use ReLU and max-pooling units in our neural network architecture, the loss function is not continuously differentiable. Nevertheless, since the set of discontinuities has

measure zero, we can assume that this will not be an issue in practice, as we will never encounter the problematic points.

Another technical issue is that, due to the not concavity of the inner problem, we are not able to compute global maximizers, since PGD will converge to local maxima. In such cases, we can consider a subset $\mathcal{S}'$ of $\mathcal{S}$ such that the local maximum is a global maximum in the region $\mathcal{S}'$. Applying the theorem for $\mathcal{S}'$ gives us that the gradient corresponds to a descent direction for the saddle point problem when the adversary is constrained in $\mathcal{S}'$. Therefore if the inner maximum is a true adversarial example for the network, then SGD using the gradient at that point will decrease the loss value at this particular adversarial examples, thus making progress towards a robust model.

These arguments suggest that the conclusions of the theorem are still valid in our saddle point problem, and –as our experiments confirm– we can solve it reliably.

## D    TRANSFERABILITY

A lot of recent literature on adversarial training discusses the phenomenon of transferability Goodfellow et al. (2014); Kurakin et al. (2016); Tramèr et al. (2017b), i.e. adversarial examples transfer between differently trained networks. This raises concerns for practical applications, since it suggests that deep networks are extremely vulnerable to attacks, *even when there is no direct access to the target network*.

This phenomenon is further confirmed by our current experiments. [4] Moreover, we notice that the extent to which adversarial examples transfer decreases as we increase either network capacity or the power of the adversary used for training the network. This serves as evidence for the fact that the transferability phenomenon can be alleviated by using high capacity networks in conjunction with strong oracles for the inner optimization problem.

**MNIST.**    In an attempt to understand these phenomena we inspect the loss functions corresponding to the trained models we used for testing transferability. More precisely, we compute angles between gradients of the loss functions evaluated over a large set of input examples, and plot their distribution. Similarly, we plot the value of the loss functions between clean and perturbed examples for both the source and transfer networks. In Figure 8 we plot our experimental findings on the MNIST dataset for $\varepsilon = 0.3$. We consider a naturally trained large network (two convolutional layers of sizes 32 and 64, and a fully connected layer of size 1024), which we train twice starting with different initializations. We plot the distribution of angles between gradients for the same test image in the two resulting networks (orange histograms), noting that they are somewhat correlated. As opposed to this, we see that pairs of gradients for random pairs of inputs for one architecture are as uncorrelated as they can be (blue histograms), since the distribution of their angles looks Gaussian.

Next, we run the same experiment on a naturally trained very large network (two convolutional layers of sizes 64 and 128, and a fully connected layer of size 1024). We notice a mild increase in classification accuracy for transferred examples.

Finally, we repeat the same set of experiments, after training the large and very large networks against the FGSM adversary. We notice that gradients between the two architectures become significantly less correlated. Also, the classification accuracy for transferred examples increases significantly compared to the naturally trained networks.

We further plot how the value of the loss function changes when moving from the natural input towards the adversarially perturbed input (in Figure 8 we show these plots for four images in the MNIST test dataset), for each pair of networks we considered. We observe that, while for the naturally trained networks, when moving towards the perturbed point, the value of the loss function on the transfer architecture tends to start increasing soon after it starts increasing on the source architecture. In contrast, for the stronger models, the loss function on the transfer network tends to start increasing later, and less aggressively.

---

[4]Our experiments involve transferability between networks with the same architecture (potentially with layers of varying sizes), trained with the same method, but with different random initializations. The reason we consider these models rather than highly different architectures is that they are likely the worst case instances for transferability.

**CIFAR10.** For the CIFAR10 dataset, we investigate the transferability of the FGSM and PGD adversaries between our simple and wide architectures, each trained on natural, FGSM and PGD examples. Transfer accuracies for the FGSM adversary and PGD adversary between all pairs of such configurations (model + training method) with independently random weight initialization are given in tables 3 and 4 respectively. The results exhibit the following trends:

- **Stronger adversaries decrease transferability:** In particular, transfer attacks between two PGD-trained models are less successful than transfer attacks between their naturally-trained counterparts. Moreover, adding PGD training helps with transferability from all adversarial datasets, *except* for those with source a PGD-trained model themselves. This applies to both FGSM attacks and PGD attacks.

- **Capacity decreases transferability:** In particular, transfer attacks between two PGD-trained wide networks are less successful than transfer attacks between their simple PGD-trained counterparts. Moreover, with few close exceptions, changing the architecture from simple to wide (and keeping the training method the same) helps with transferability from all adversarial datasets.

We additionally plotted how the loss of a network behaves in the direction of FGSM and PGD examples obtained from itself and an independently trained copy; results for the simple naturally trained network and the wide PGD trained network are given in Table 7. As expected, we observe the following phenomena:

- sometimes, the FGSM adversary manages to increase loss faster near the natural example, but as we move towards the boundary of the $\ell_\infty$ box of radius $\varepsilon$, the PGD attack always achieves higher loss.

- the transferred attacks do worse than their white-box counterparts in terms of increasing the loss;

- and yet, the transferred PGD attacks dominate the white-box FGSM attacks for the naturally trained network (and sometimes for the PGD-trained one too).

Table 3: CIFAR10: black-box FGSM attacks. We create FGSM adversarial examples with $\varepsilon = 8$ from the evaluation set on the source network, and then evaluate them on an independently initialized target network.

| Source \ Target | Simple (natural training) | Simple (FGSM training) | Simple (PGD training) | Wide (natural training) | Wide (FGSM training) | Wide (PGD training) |
|---|---|---|---|---|---|---|
| Simple (natural training) | 32.9% | 74.0% | 73.7% | 27.6% | 71.8% | 76.6% |
| Simple (FGSM training) | 64.2% | 90.7% | 60.9% | 61.5% | 90.2% | 67.3% |
| Simple (PGD training) | 77.1% | 78.1% | 60.2% | 77.0% | 77.9% | 66.3% |
| Wide (natural training) | 34.9% | 78.7% | 80.2% | 21.3% | 75.8% | 80.6% |
| Wide (FGSM training) | 64.5% | 93.6% | 69.1% | 53.7% | 92.2% | 72.8% |
| Wide (PGD training) | 85.8% | 86.6% | 73.3% | 85.6% | 86.2% | 67.0% |

Table 4: CIFAR10: black-box PGD attacks. We create PGD adversarial examples with $\varepsilon = 8$ for 7 iterations from the evaluation set on the source network, and then evaluate them on an independently initialized target network.

| Target \ Source | Simple (natural training) | Simple (FGSM training) | Simple (PGD training) | Wide (natural training) | Wide (FGSM training) | Wide (PGD training) |
|---|---|---|---|---|---|---|
| Simple (natural training) | 6.6% | 71.6% | 71.8% | 1.4% | 51.4% | 75.6% |
| Simple (FGSM training) | 66.3% | 40.3% | 58.4% | 65.4% | 26.8% | 66.2% |
| Simple (PGD training) | 78.1% | 78.2% | 57.7% | 77.9% | 78.1% | 65.2% |
| Wide (natural training) | 10.9% | 79.6% | 79.1% | 0.0% | 51.3% | 79.7% |
| Wide (FGSM training) | 67.6% | 51.7% | 67.4% | 56.5% | 0.0% | 71.6% |
| Wide (PGD training) | 86.4% | 86.8% | 72.1% | 86.0% | 86.3% | 64.2% |

Table 5: CIFAR10: white-box attacks for $\varepsilon = 8$. For each architecture and training method, we list the accuracy of the resulting network on the full CIFAR10 evaluation set of 10,000 examples. The FGSM random method is the one suggested by Tramèr et al. (2017a), whereby we first do a small random perturbation of the natural example, and the apply FGSM to that.

| Model \ Adversary | Natural | FGSM | FGSM random | PGD (7 steps) | PGD (20 steps) |
|---|---|---|---|---|---|
| Simple (natural training) | 92.7% | 27.5% | 19.6% | 1.2% | 0.8% |
| Simple (FGSM training) | 87.4% | 90.9% | 90.4% | 0.0% | 0.0% |
| Simple (PGD training) | 79.4% | 51.7% | 55.9% | 47.1% | 43.7% |
| Wide (natural training) | 95.2% | 32.7% | 25.1% | 4.1% | 3.5% |
| Wide (FGSM training) | 90.3% | 95.1% | 95.0% | 0.0% | 0.0% |
| Wide (PGD training) | 87.3% | 56.1% | 60.3% | 50.0% | 45.8% |

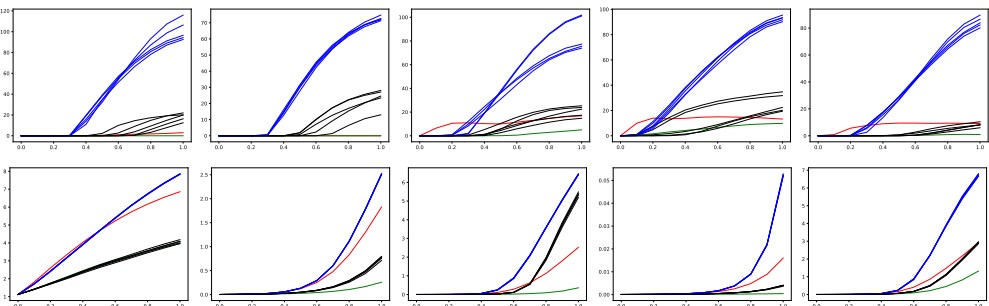

Figure 7: CIFAR10: change of loss function in the direction of white-box and black-box FGSM and PGD examples with $\varepsilon = 8$ for the same five natural examples. Each line shows how the loss changes as we move from the natural example to the corresponding adversarial example. Top: simple naturally trained model. Bottom: wide PGD trained model. We plot the loss of the original network in the direction of the FGSM example for the original network (red lines), 5 PGD examples for the original network obtained from 5 random starting points (blue lines), the FGSM example for an independently trained copy network (green lines) and 5 PGD examples for the copy network obtained from 5 random starting points (black lines). All PGD attacks use 100 steps with step size 0.3.

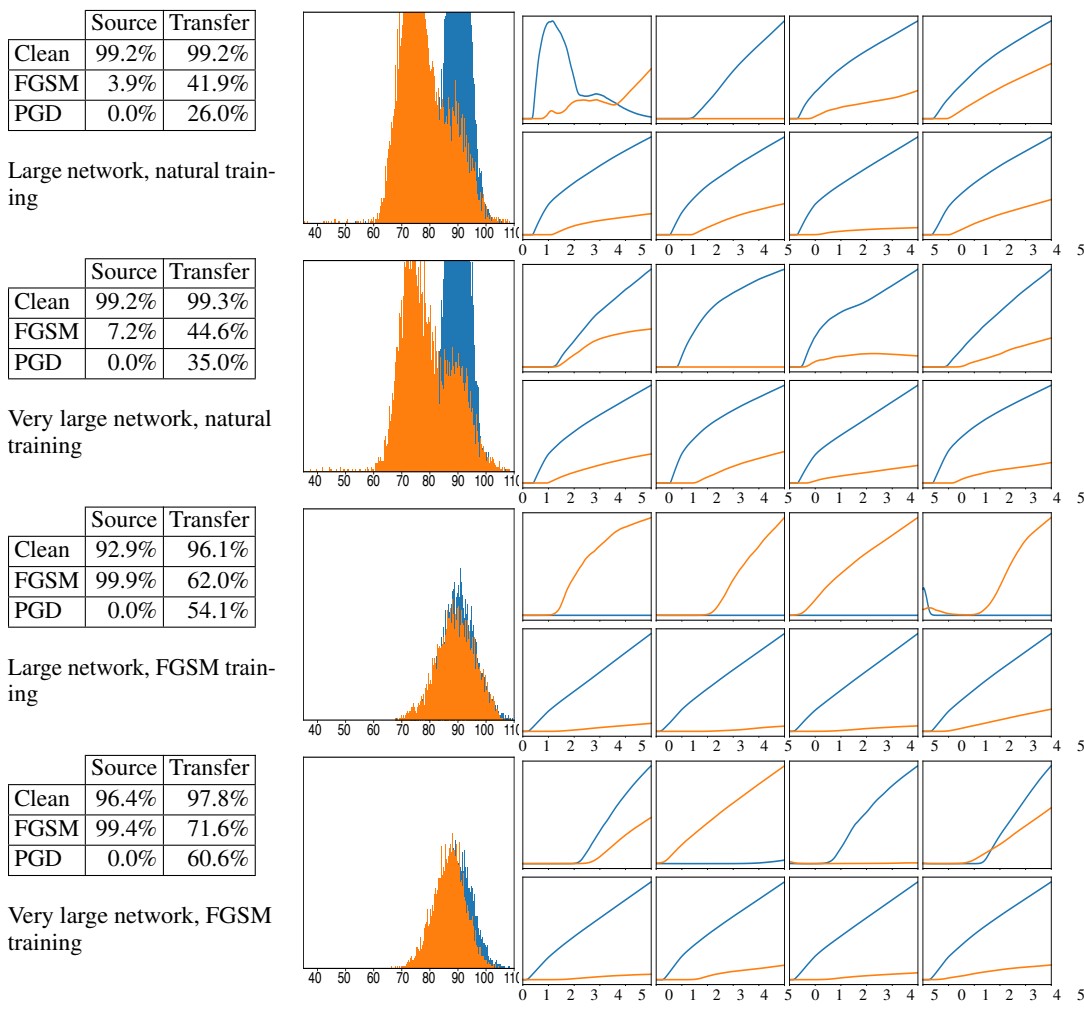

|        | Source | Transfer |
|--------|--------|----------|
| Clean  | 99.2%  | 99.2%    |
| FGSM   | 3.9%   | 41.9%    |
| PGD    | 0.0%   | 26.0%    |

Large network, natural training

|        | Source | Transfer |
|--------|--------|----------|
| Clean  | 99.2%  | 99.3%    |
| FGSM   | 7.2%   | 44.6%    |
| PGD    | 0.0%   | 35.0%    |

Very large network, natural training

|        | Source | Transfer |
|--------|--------|----------|
| Clean  | 92.9%  | 96.1%    |
| FGSM   | 99.9%  | 62.0%    |
| PGD    | 0.0%   | 54.1%    |

Large network, FGSM training

|        | Source | Transfer |
|--------|--------|----------|
| Clean  | 96.4%  | 97.8%    |
| FGSM   | 99.4%  | 71.6%    |
| PGD    | 0.0%   | 60.6%    |

Very large network, FGSM training

Figure 8: Transferability experiments for four different instances (naturally trained large and very large networks, and FGSM-trained large and very large networks, respectively). For each instance we ran the same training algorithm twice, starting from different initializations. Tables on the left show the accuracy of the networks against three types of input (clean, perturbed with FGSM, perturbed with PGD ran for 40 steps); the first column shows the resilience of the first network against examples produced using its own gradients, the second column shows resilience of the second network against examples transferred from the former network. The histograms reflect angles between pairs of gradients corresponding to the same inputs versus the baseline consisting of angles between gradients from random pairs of points. Images on the right hand side reflect how the loss functions of the native and the transfer network change when moving in the direction of the perturbation; the perturbation is at 1 on the horizontal axis. Plots in the top row are for FGSM perturbations, plots in the bottom row are for PGD perturbations produced over 40 iterations.

# E MNIST INSPECTION

The robust MNIST model described so far is small enough that we can visually inspect most of its parameters. Doing so will allow us to understand how it is different from a naturally trained variant and what are the general characteristics of a network that is robust against $\ell_\infty$ adversaries. We will compare three different networks: a naturally trained model, and two adversarially trained ones. The latter two models are identical, modulo the random weight initialization, and were used as the public and secret models used for our robustness challenge.

Initially, we examine the first convolutional layer of each network. We observe that the robust models only utilize 3 out of the total 32 filters, and for each of these filters only one weight is non-zero. By doing so, the convolution degrades into a scaling of the original image. Combined with the bias and the ReLU that follows, this results in a *thresholding filter*, or equivalently ReLU($\alpha x - \beta$) for some constants $\alpha$, $\beta$. From the perspective of adversarial robustness, thresholding filters are immune to any perturbations on pixels with value less than $\beta - \varepsilon$. We visualize a sample of the filters in Figure 9 (plots a, c, and e).

Having observed that the first layer of the network essentially maps the original image to three copies thresholded at different values, we examine the second convolutional layer of the classifier. Again, the filter weights are relatively sparse and have a significantly wider value range than the naturally trained version. Since only three channels coming out of the first layer matter, is follows (and is verified) that the only relevant convolutional filters are those that interact with these three channels. We visualize a sample of the filters in Figure 9 (plots b, d, and f).

Finally, we examine the softmax/output layer of the network. While the weights seem to be roughly similar between all three version of the network, we notice a significant difference in the class biases. The adversarially trained networks heavily utilize class biases (far from uniform), and do so in a way very similar to each other. A plausible explanation is that certain classes tend to be very vulnerable to adversarial perturbations, and the network learns to be more conservative in predicting them. The plots can be found in Figure 10.

All of the "tricks" described so far seem intuitive to a human and would seem reasonable directions when trying to increase the adversarial robustness of a classifier. We emphasize the none of these modifications were hard-coded in any way and they were all *learned solely through adversarial training*. We attempted to manually introduce these modifications ourselves, aiming to achieve adversarial robustness without adversarial training, but with no success. A simple PGD adversary could fool the resulting models on all the test set examples.

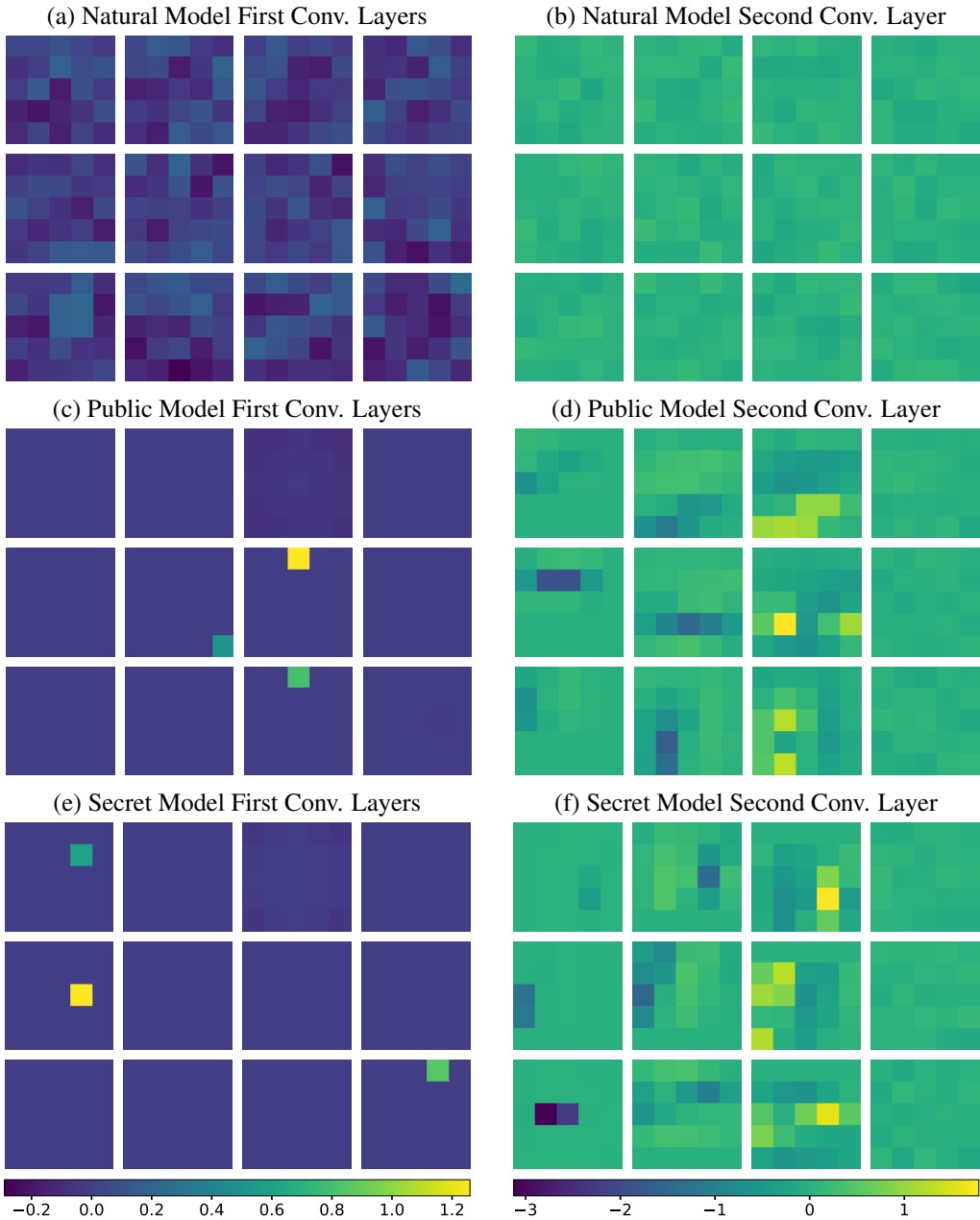

Figure 9: Visualizing a sample of the convolutional filters. For the natural model (a,b) we visualize random filters, since there is no observable difference in any of them. For the first layer of robust networks we make sure to include the 3 non-zero filters. For the second layer, the first three columns represent convolutional filters that utilize the 3 non-zero channels, and we choose the most interesting ones (larger range of values). We observe that adversarially trained networks have significantly more concentrated weights. Moreover, the first convolutional layer degrades into a few thresholding filters.

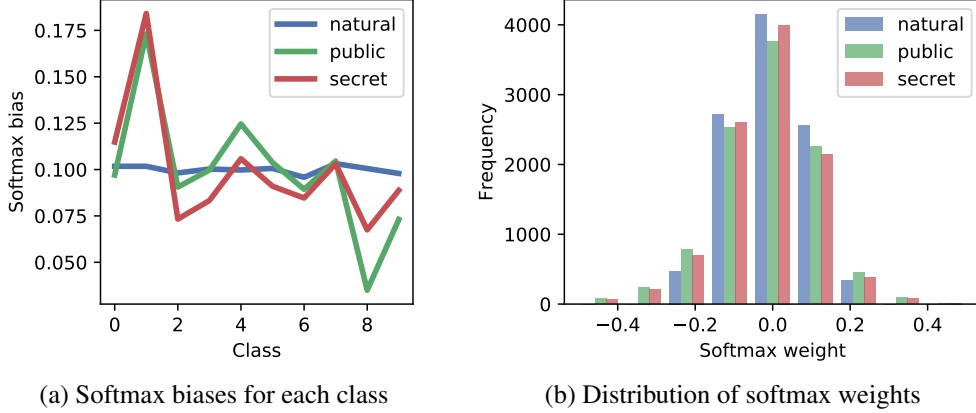

(a) Softmax biases for each class

(b) Distribution of softmax weights

Figure 10: Softmax layer examination. For each network we create a histogram of the layer's weights and plot the per-class bias. We observe that while weights are similar (slightly more concentrated for the natural one) the biases are far from uniform and with a similar pattern for the two adversarially trained networks.

## F SUPPLEMENTARY FIGURES

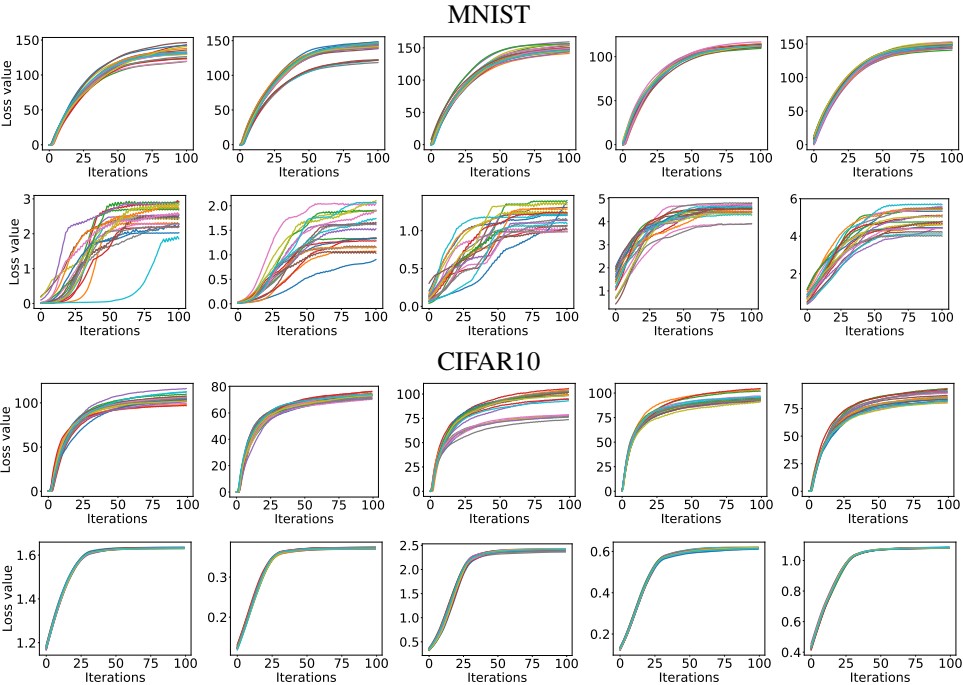

Figure 11: Loss function value over PGD iterations for 20 random restarts on random examples. The 1st and 3rd rows correspond to naturally trained networks, while the 2nd and 4th to adversarially trained ones.

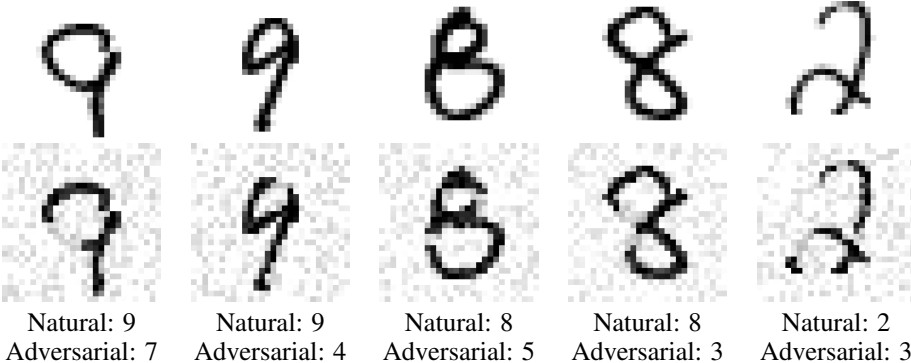

Figure 12: Sample adversarial examples with $\ell_2$ norm bounded by $4$. The perturbations are significant enough to cause misclassification by humans too.

