# OpenReview forum: "Towards Deep Learning Models Resistant to Adversarial Attacks"
_ICLR.cc/2018/Conference — Accept (Poster)_

### Official Review · AnonReviewer1 · 2017-11-26

**Rating:** 6
**Confidence:** 3

**Review:**

This paper proposes to look at making neural networks resistant to adversarial loss through the framework of saddle-point problems. They show that, on MNIST, a PGD adversary fits this framework and allows the authors to train very robust models. They also show encouraging results for robust CIFAR-10 models, but with still much room for improvement. Finally, they suggest that PGD is an optimal first order adversary, and leads to optimal robustness against any first order attack.

This paper is well written, brings new ideas and perfoms interesting experiments, but its claims are somewhat bothering me, considering that e.g. your CIFAR-10 results are somewhat underwhelming. All you've really proven is that PGD on MNIST seems to be the ultimate adversary. You contrast this to the fact that the optimization is non-convex, but we know for a fact that MNIST is fairly simple in that regime; iirc a linear classifier gets something like 91% accuracy on MNIST. So my guess is that the optimization problem on MNIST is in fact pretty convex and mostly respects the assumptions of Danskin's theorem, but not so much for CIFAR-10 (maybe even less so for e.g. ImageNet, which is what Kurakin et al. seem to find).

Considering your CIFAR-10 results, I don't think anyone should "suggest that secure neural networks are within reach", because 1) there is still room for improvement 2) it's a safe bet that someone will always just come up with a better attack than whatever defense we have now. It has been this way in many disciplines (crypto, security) for centuries, I don't see why deep learning should be exempt. Simply saying "we believe that our robust models are significant progress on the defense side" was enough, because afaik you did improve on CIFAR-10's SOTA; don't overclaim.
You make these kinds of claims in a few other places in this paper, please be careful with that.

The contributions in your appendix are interesting.
Appendix A somewhat confirms one of the postulates in Goodfellow et al. (2014): "The direction of perturbation, rather than the specific point in space, matters most. Space is not full of pockets of adversarial examples that finely tile the reals like the rational numbers".
Appendix B and C are not extremely novel in my mind, but definitely add more evidence.
Appendix E is quite nice since it gives an insight into what actually makes the model resistant to adversarial examples.


Remarks:
- The update for PGD should be using \nabla_{x_t} L(\theta,x_t,y), (rather than only \nabla_x)?
- In table 2, attacking a with 20-step PGD is doing better than 7-step.  When you say "other hyperparameter choices didn’t offer a significant decrease in accuracy", does that include the number of steps? If not why stop there? What happens for more steps? (or is it too computationally intensive?)
- You only seem to consider adversarial examples created from your dataset + adv. noise. What about rubbish class examples? (e.g. rgb noise)

---

> ### Author Response · Authors · 2018-01-05
> **Review Response**
>
> We thanks the reviewer for providing feedback.
>
> Regarding "secure networks are within reach" claim: We definitely agree that there is (large) room for improvement on CIFAR10. Our claim, however, comes from the fact that (to the best of our knowledge) the classifiers we trained were the first ones to robustly classify any non-trivial fraction of the test set. We indeed believe (and provide experimental evidence for it) that no attack will significantly decrease the accuracy of our classifier. We view our results as a baseline that shows that classification robustness is indeed achievable (against a well-defined class of adversaries at least). We agree that our claims ended up sounding too strong though. We will update our paper to tone them down.
>
> Regarding CIFAR10 results: The reviewer points out that the optimization landscape for MNIST is much simpler than that of CIFAR10 and that this would  explain the difference in the performance of the resulting classifiers. We want to point out, however, that our performance on CIFAR10 is due to poor generalization and not the difficulty of the training problem itself. As can be seen in Figure 1b, we are able to train a perfectly robust classifier with 100% adversarial accuracy on the training set. This shows that the optimization landscape of the problem is still tractable with a PGD adversary.
>
> Regarding CIFAR10 attack parameters: We didn’t explore additional parameters due to the computational constraints at the time. Overall we have observed that the number of PGD steps does not change the resulting accuracy by more than a few percent. For instance, if we retrain the (non-wide version of the) CIFAR10 network with a 5-step PGD adversary we get the following accuracies when testing against PGD: 5 steps -> 45.00%, 10 steps -> 43.02%, 20 steps -> 42.65%, 100 steps ->  42.21%.
>
> Regarding rubbish class examples: We agree that rubbish class examples are an important class to consider. However it is unclear how to rigorously define them. As we discuss in our paper (3rd paragraph of page 3), providing any kind of robustness guarantees requires a precise definitions of the allowed adversarial perturbations.

---

### Official Review · AnonReviewer2 · 2017-11-27
**Interesting experimental results, but insufficient to support some of the strong claims made in the paper**

**Rating:** 7
**Confidence:** 4

**Review:**

- The authors investigate a minimax formulation of deep network learning to increase their robustness, using projected gradient descent as the main adversary. The idea of formulating the threat model as the inner maximization problem is an old one. Many previous works on dealing with uncertain inputs in classification apply this minimax approach using robust optimization, e.g.:

https://www2.eecs.berkeley.edu/Pubs/TechRpts/2003/CSD-03-1279.pdf
http://www.jmlr.org/papers/volume13/ben-tal12a/ben-tal12a.pdf

In the case of convex uncertainty sets, many of these problems can be solved efficiently to a global minimum. Generalization bounds on the adversarial losses can also be proved. Generalizing this approach to non-convex neural network learning makes sense, even when it is hard to obtain any theoretical guarantees.

- The main novelty is the use of projected gradient descent (PGD) as the adversary. From the experiments it seems training with PGD is very robust against a set of adversaries including fast gradient sign method (FGSM), and the method proposed in Carlini & Wagner (CW). Although the empirical results are promising, in my opinion they are not sufficient to support the bold claim that PGD is a 'universal' first order adversary (on p2, in the contribution list) and provides broad security guarantee (in the abstract). For example, other adversarial example generation methods such as DeepFool and Jacobian-based Saliency Map approach are missing from the comparison. Also it is not robust to generalize from two datasets MNIST and CIFAR alone.

- Another potential issue with using projected gradient descent as adversary is the quality of the adversarial example generated. The authors show empirically that PGD finds adversarial examples with very similar loss values on multiple runs. But this does not exclude the possibility that PGD with different step sizes or line search procedure, or the use of randomization strategies such as annealing, can find better adversarial examples under the same threat model. This could make the robustness of the network rather dependent on the specific implementation of PGD for the inner maximization problem.

- In Tables 3, 4, and 5 in the appendix, in most cases models trained with PGD are more robust than models trained with FGSM as adversary, modulo the phenomenon of label leakage when using FGSM as attack. However in the bottom right corner of Table 4, FGSM training seems to be more robust than PGD training against black box PGD attacks. This raises the question on whether PGD is truly 'universal' and provides broad security guarantees, once we add more first order attacks methods to the mix.

---

> ### Author Response · Authors · 2018-01-05
> **Review Response**
>
> We thank the reviewer for the feedback.
>
> We agree with the reviewer that min-max approach for robust classification have been studied before. As we mention in our paper (right after equation 2.1), such formulations go back at least to the work of Abraham Wald in the 1940s (e.g., see https://www.jstor.org/stable/1969022). What we view as the main contribution of our paper lies, however, not in introducing a new problem formulation but in studying if such formulation can inform training methods that lead to reliably robust deep learning models in practice.
>
> We do not claim that training with a PGD adversary is the main novelty of our paper - prior work has already employed a variety of iterative first-order methods. Instead, our goal is to argue that training with PGD is a principled approach to adversarial robustness and to give both theoretical and empirical evidence for this view. (See the connection via Danskin’s theorem, and our loss function explorations in Appendix A.) Moreover, we demonstrate that adversarial training with PGD - when done properly - leads to state-of-the-art robustness on two canonical datasets. In contrast to much other work in this area, we have also validated the robustness of our models via a public challenge in which our model underwent (unsuccessful) attacks by other research groups.
>
> Regarding PGD being a "universal" first-order adversary and the "broad security guarantee" claim: first, we would like to note that on Page 2 of our paper, we state that we provide evidence for this view, not that this view is necessarily correct. Still, we believe that from the point of view of first order methods, our evaluation approach is comprehensive. Moreover, it is also worth noting that there has been increasing evidence for this view since we first published our paper: (i) No researcher has been able to break our released models. (ii) Follow-up work has used verification tools to test the PGD approaches and found that the adversarial examples found by iterative first order methods are almost as good as the adversarial examples found with a computationally expensive exhaustive search. Hence we believe that at least in the context of L_infinity robustness, viewing PGD as a universal first-order adversary has merit.
>
> Regarding JSMA and DeepFool: JSMA is an attack that is designed to perturb as few pixels as possible (often by a large value). Restricting this attack to the space of attacks we consider (0.3 distance from original in L_infinity norm) leads to an attack that is very slow and, as far as we could tell, less potent than PGD. Deepfool is an attack that has been designed with an aim of computing minimum norm perturbations. For the regime we are studying, the only difference between DeepFool and the CW attack is the choice of target class to use at each step. We didn’t feel that testing against this variation was necessary (given the length of our paper). Again we want to emphasize that we invited the community to attempt attacks against our published model and we didn’t receive any attacks that significantly lowered the performance of our model. Nevertheless we will perform the suggested experiments and add them to the final version.
>
> Regarding the point about PGD step size and variations: While one needs to tune PGD to a certain degree, we found that the method is robust to reasonable changes in the choice of PGD parameters. Training against different PGD variants also leads to robust networks.
>
> Regarding Table 4: We emphasize that Table 4 contains results for transfer attacks, which add an additional complication due to the mismatch between the source model used to construct the attack, and the target model that we would like to attack. We do observe that training with FGSM offers more robustness against *transfer* attacks constructed using networks trained with PGD. There is an important caveat however. The larger robustness is due to the difference in the two models, not because FGSM produces inherently more robust models. We view this effect as an artifact of the transferability phenomenon rather than a fundamental shortcoming of PGD-based adversarial training. When we consider the minimum across all columns in each row, the PGD-trained target model offers significantly more robustness than the FGSM-trained model (64.2% vs. 0.0%).

---

### Official Review · AnonReviewer3 · 2017-11-28
**The unreasonable effectiveness of gradient descent**

**Rating:** 7
**Confidence:** 4

**Review:**

This paper consolidates and builds on recent work on adversarial examples and adversarial training for image classification. Its contributions:

 - Making the connection between adversarial training and robust optimization more explicit.

 - Empirical evidence that:
   * Projected gradient descent (PGD) (as proposed by Kurakin et al. (2016)) reasonably approximates the optimal attack against deep convolutional neural networks
   * PGD finds better adversarial examples, and training with it yields more robust models, compared to FGSM

 - Additional empirical analysis:
   * Comparison of weights in robust and non-robust MNIST classifiers
   * Vulnerability of L_infty-robust models to to L_2-bounded attacks

The evidence that PGD consistently finds good examples is fairly compelling -- when initialized from 10,000 random points near the example to be disguised, it usually finds examples of similar quality. The remaining variance that's present in those distributions shouldn't hurt learning much, as long as a significant fraction of the adversarial examples are close enough to optimal.

Given the consistent effectiveness of PGD, using PGD for adversarial training should yield models that are reliably robust (for a specific definition of robustness, such as bounded L_infinity norm). This is an improvement over purely heuristic approaches, which are often less robust than claimed.

The comparison to R+FGSM is interesting, and could be extended in a few small ways. What would R+FGSM look like with 10,000 restarts? The distribution should be much broader, which would further demonstrate how PGD works better on these models. Also, when generating adversarial examples for testing, how well would R+FGSM work if you took the best of 2,000 random restarts? This would match the number of gradient computations required by PGD with 100 steps and 20 restarts. Again, I expect that PGD would be better, but this would make that point clearer. I think this analysis would make the paper stronger, but I don't think it's required for acceptance, especially since R+FGSM itself is such a recent development.

One thing not discussed is the high computational cost: performing a 40-step optimization of each training example will be ~40 times slower than standard stochastic gradient descent. I suspect this is the reason why there are results on MNIST and CIFAR, but not ImageNet. It would be very helpful to add some discussion of this.

The title seems unnecessarily vague, since many papers have been written with the same goal -- make deep learning models resistant to adversarial attacks. (This comment does not affect my opinion about whether or not the paper should be accepted, and is merely a suggestion for the authors.)

Also, much of the paper's content is in the appendices. This reads like a journal article where the references were put in the middle. I don't know if that's fixable, given conference constraints.

---

> ### Author Response · Authors · 2018-01-05
> **Review Response**
>
> We thank the reviewer for the positive feedback.
>
> Regarding R+FGSM: We evaluated our robust networks against R+FGSM with multiple restarts and got the following results.
> - MNIST. PGD-40: 93.2%, R+FGSM x40: 92.2%, R+FGSM x2000: 90.51%.
> - CIFAR10 (non-wide). PGD-10: 43.02%, R+FGSM x10: 50.17%, R+FGSM x2000: 48.66%.
> These experiments suggest that for evaluation purposes R+FGSM is qualitatively similar to PGD (at least for adversarially trained networks). Still if one attempts to adversarially *train* using R+FGSM, the resulting classifier overfits to the R+FGSM perturbations and while achieving high training and test accuracy against R+FGSM, it is completely vulnerable to PGD.
>
> We also created loss histograms to compare PGD and R+FGSM with the results plotted at https://ibb.co/gcJuxG (final loss value frequency over 10,000 random restarts for 5 random examples). We observe that R+FGSM exhibits a similar concentration phenomenon to that observed for PGD in Appendix A. We will include these experiments in the final paper version. We agree that further investigating the difference between PGD and R+FGSM is worth exploring in subsequent research.
>
> Regarding the computational cost of robust optimization: It is indeed true that training against a PGD adversary increases the training time by a factor that is roughly equal to the number of PGD steps. This is a drawback of this method that we hope will be addressed in future research. But, at this point, getting sufficient robustness indeed leads to a running time overhead.  We will add a brief discussion of this in the final version of the paper.
>
> Regarding title choice: Our intention was to convey that there exist robustness baselines that current techniques can achieve. Still, we agree that the title might be too vague and we will revisit our choice.

---

### Public Comment · (anonymous) · 2017-11-10
**Certified Defenses for Data Poisoning Attacks**

This NIPS paper seems to have the same formulation for defending against attacks: https://arxiv.org/pdf/1706.03691.pdf

They consider the same min-max formulation and try to build a defense against attacks (similar idea to current paper?).

They consider data-poisoning, but one could imagine the same ideas applied to adversarial attacks. Does the current paper propose something very new from what was described in the NIPS paper? I'm not a specialist in analysis or large-scale optimization, hence my question.

Thanks!

---

> ### Author Response · Authors · 2017-11-10
> **Re: Certified Defenses for Data Poisoning Attacks**
>
> We thank the reviewer for inquiring about the novelty of our work. As we point out in our paper (see Page 3), the min-max formulation itself is not new. In fact, problem formulations of this form have been studied for multiple decades (c.f. the work of Abraham Wald). Moreover, there is a rich literature concerning min-max problems in robust optimization. Claiming a min-max formulation as new would ignore a significant body of prior work.
>
> As we point out in the introduction, our main contribution is *how* we employ the min-max formulation to study adversarially robust machine learning. To the best of our knowledge, our paper is the first detailed study of the min-max formulation for robust neural networks. From a scientific point of view, the question is not only whether adversarial robustness can be described with a min-max formulation, but whether such a formulation actually matches the computational reality we face in the practice of deep learning.
>
> One concrete contribution is our thorough experiments exploring the adversarial loss landscape. Combined with Danskin’s theorem, they give evidence to the theory that adversarial training is indeed a principled way to solve the aforementioned min-max problem. Furthermore, our paper conducted the first public attack challenge to ascertain the robustness of a proposed defense. The challenge showed that our MNIST model is the first deep network that could not easily be broken with a new attack (subject to l_infinity constraints).
>
> Finally, we would also like to point out that our paper appeared publicly nearly concurrently with the NIPS paper mentioned by the reviewer. Moreover, there are several important differences compared to this paper. For instance, their focus is on robustness to corrupt training data, while our paper is about robustly classifying new unseen examples. Overall, we believe that reducing the comparison with the cited NIPS paper to the min-max formulation is an oversimplification of both their work and our work.

---

> > ### Public Comment · ~Jacob_Steinhardt1 · 2017-11-17
> > **Re: re: data poisoning paper**
> >
> > As the author of the data poisoning paper that is mentioned, I just wanted to agree with the authors of the current paper that these papers (in my opinion) are quite different. It is of course interesting to compare the similarities/differences in the min-max formulation, but the problems studied in the two papers are so different that this would (again, in my opinion) have no affect whatsoever on novelty of the current submission.

---

### Public Comment · (anonymous) · 2017-11-28
**Lyu et al. (2015)**

Lyu et al. (2015) has made the connection to robust optimization (and proposed a new regularization).
Please cite their work https://arxiv.org/pdf/1511.06385.pdf when you introduce the minimax formulation.

Having said this, I understand that the main contribution of this paper is a systematic empirical study of the minimax formulation. The abstract and the intro seem to give the impression that this is the first paper that suggests a unifying minimax framework. Citing other works and attributing credit accordingly would help emphasize the true contributions of the paper.

---

> ### Author Response · Authors · 2017-11-29
> **Re: Lyu et al. (2015)**
>
> We thank the reviewer for bringing the work of Lyu et al. to our attention. We will cite it and discuss it in the "Related Work" section of our updated paper. As we already point out in our paper, both the general min-max framework, as well as its application to the problem of adversarial examples, are not new. Min-max formulations have been used extensively in the context of robust optimization and statistics, going back at least to the work of Abraham Wald in the 1930s and 40s. In the context of adversarial examples, we already cite the work of Shaham et al. (https://arxiv.org/abs/1511.05432) and Huang et al. (https://arxiv.org/abs/1511.03034), which consider a similar min-max formulation and appeared on arXiv nearly concurrently with the work of Lyu et al.
>
> To clarify our contributions: the min-max formulation is part of the approach and *not* claimed as a contribution (see our introduction and the reply to "Certified Defenses for Data Poisoning Attacks" above). Instead, one of our main contributions is to study the loss landscape of the saddle point problem, *without replacing the loss by its first-order approximation*. It is known that solving the saddle point problem with a first-order approximation of the loss (see Figure 6 of Appendix B in our paper) produces networks that are vulnerable to more sophisticated (multi-step) attacks.

---

### Public Comment · ~Nicholas_Carlini1 · 2018-01-23
**Our analysis of this paper**

We have been performing an analysis of the robustness of many of the papers submitted here. This paper provides a substantially stronger defense than many of the other submissions, and we were not able to meaningfully invalidate any of the claims made. Given our analysis so far, it looks like this is the strongest defense submitted to ICLR 2018.

---

### Decision · Program_Chairs · 2018-01-29
**ICLR 2018 Conference Acceptance Decision**

**Decision:**

Accept (Poster)

**Comment:**

This paper presents new results on adversarial training, using the framework of robust optimization. Its minimax nature allows for principled methods of both training and attacking neural networks.

The reviewers were generally positive about its contributions, despite some concerns about 'overclaiming'. The AC recommends acceptance, and encourages the authors to also relate this work with the concurrent ICLR submission (https://openreview.net/forum?id=Hk6kPgZA-) which addresses the problem using a similar approach.

---

> ### Public Comment · ~Nicholas_Carlini1 · 2018-02-02
> **Regarding "overclaiming"**
>
> We have done a security analysis of this defense, and we were not able to decrease the robustness of the classifier within the threat model considered: our best attacks succeeded in reducing classifier accuracy to 47%. This is in contrast to 6 (of 8 total defenses that argue white-box security and are non-certified) defenses accepted to ICLR 2018, which we break completely, and 1 that we can reduce the efficacy of but don't completely break. We believe that this paper does not overclaim, as suggested by the reviews: in fact, it’s the only paper that has held up to its claims. See our paper for details: http://arxiv.org/abs/1802.00420.